# Nutritional Status, Sex, and Ambient Temperature Modulate the Wingbeat Frequency of the Diamondback Moth *Plutella xylostella*

**DOI:** 10.3390/insects15020138

**Published:** 2024-02-19

**Authors:** Menglun Wang, Jialin Wang, Pei Liang, Kongming Wu

**Affiliations:** 1Institute for the Control of Agrochemicals, Ministry of Agriculture and Rural Affairs of the People’s Republic of China, Beijing 100125, China; wangmenglun@126.com; 2Guangdong Laboratory for Lingnan Modern Agriculture, Guangzhou 510642, China; 3National Key Laboratory of Intelligent Tracking and Forecasting for Infectious Diseases, National Institute for Communicable Disease Control and Prevention, Chinese Center for Disease Control and Prevention, Beijing 102206, China; 4Department of Entomology, China Agricultural University, Beijing 100193, China; 5State Key Laboratory for Biology of Plant Diseases and Insect Pests, Institute of Plant Protection, Chinese Academy of Agricultural Sciences, Beijing 100193, China

**Keywords:** *Plutella xylostella*, wingbeat frequency, stroboscope, temperature, humidity

## Abstract

**Simple Summary:**

The diamondback moth, *Plutella xylostella* (L.), is an important migratory insect pest worldwide. In this study, the effects of environmental parameters, age, nutritional supplementation and sex on the wingbeat frequency of the insect adult were measured in the laboratory, and the results showed that nutritional status, sex and ambient temperature modulate the wingbeat frequency of the moth. These findings can be helpful for further understanding flight biology of the insect as well as to improve the accuracy using insect radar facility to directly identify migrating moths based on species difference of wingbeat frequency.

**Abstract:**

The diamondback moth, *Plutella xylostella* L. (Lepidoptera: Plutellidae), is a cosmopolitan horticultural pest that is undergoing a fast, climate-driven range expansion. Its wide geographic distribution, pest status, and high incidence of insecticide resistance are directly tied to long-distance migration. Wingbeat frequency (WBF) is a key aspect of *P. xylostella* migratory behavior, but has received limited scientific attention. Here, we investigated the effects of environmental parameters, age, adult nutrition, and sex on *P. xylostella* WBF. Across experimental regimes, WBF ranged from 31.39 Hz to 78.87 Hz. Over a 10–35 °C range, the WBF of both male and female moths increased with temperature up to 62.96 Hz. Though male WBF was unaffected by humidity, females exhibited the highest WBF at 15% relative humidity (RH). WBF was unaffected by adult age, but adult nutrition exerted important impacts. Specifically, the WBF of moths fed honey water (54.66 Hz) was higher than that of water-fed individuals (49.42 Hz). Lastly, males consistently exhibited a higher WBF than females. By uncovering the biological and (nutritional) ecological determinants of diamondback moth flight, our work provides invaluable guidance to radar-based monitoring, migration forecasting, and the targeted deployment of preventative mitigation tactics.

## 1. Introduction

The diamondback moth (DBM), *Plutella xylostella* (L.), is a globally important horticultural pest that engages in long-range migration [1]. Considered to be the most widely distributed lepidopteran pest [1], DBM occurs and reproduces in all continents except Antarctica [1,2]. DBM primarily affects cruciferous vegetables such as cabbage, cauliflower, mustard, cabbage, radish, and rapeseed [1]. Every year, *P. xylostella*-induced crop losses amount to USD 4–5 billion globally [3] and USD 770 million in China [4]. Hence, the design and implementation of reliable, effective, and economically sound prevention and control measures is of critical importance. Though DBM outbreaks have been historically restricted to southern China, DBM is increasingly posing severe problems in northern parts of the country, an issue that has been ascribed to global warming and climate-driven migration [5].

Many insects engage in migration to adapt to seasonal changes in the environment or ephemeral food sources [6]. This involves an annually recurring, cross-latitude movement of large numbers of multiple species, including crop pests [7,8,9,10,11]. The in-depth scientific study of this climate-driven migration behavior for crop-feeding herbivores (and their associated natural enemies) can inform more sustainable forms of crop protection [12]. Specifically, knowledge of the (environmental) determinants of migration can help to provide point-specific forecasts and early warning of pest outbreaks, enabling a timely deployment of prevention and control measures [10]. Over the past decades, the migration behavior, e.g., migration trajectories and the flight biology of *P. xylostella*, has been intensely studied. Scholars have found that, in northern China, *P. xylostella* generally migrates northward from April to June and southward from September to October [9,13]. The northward migrating population is substantially larger than that of south-bound individuals [14]. Further, *P. xylostella* flight ability is influenced by various environmental and biological factors, including temperature, mating, and age [15]. However, it remains unclear how those factors impact wingbeat frequency.

Wingbeat frequency (WBF) is a key determinant of insect flight, as insects need to generate sufficient lift through the flapping of their wings to initiate flight [16]. WBF varies substantially within and among species, ranging between 6.7 Hz and 84.5 Hz for a set of 85 migratory Lepidoptera [17]. By characterizing the WBF of a given insect species, one not only gains insights into the determinants of its population survival but also enables myriad practical applications, e.g., in the pest management domain. Given the unique WBF ‘signatures’ of a given species, this parameter can be used to identify insect species during flight [18]. In recent years, radar-based recordings of WBF are increasingly employed to ascertain the identity of multiple migratory crop pests, e.g., locusts, planthoppers or Tephritid fruit flies [19,20,21,22,23]. This information can then be combined with trajectory simulation to infer insect migration routes, improve prediction accuracy, and provide early warnings for (climate-driven) pest outbreaks [19,24]. Therefore, the study of insect WBF plays a significant role in clarifying migration routes and in devising cost-effective pest mitigation strategies. Yet, given that environmental and biological factors cause intra-specific variability in WBF [25,26], they potentially can compromise prediction accuracy and the overall reliability of radar-based projections.

In this study, we clarify to what extent *P. xylostella* WBF is affected by environmental and biological factors. Drawing upon laboratory-based stroboscope recordings, we studied the effects of temperature, humidity, age, adult nutrition, and sex on WBF. Next, we combined laboratory-derived data with other known *P. xylostella* migration parameters in order to improve the recognition accuracy of insect radar. Our work can thus help to clarify DBM migration dynamics in China or abroad, allowing for a reliable prediction of pest occurrence and a timely deployment of mitigation measures.

## 2. Materials and Methods

### 2.1. Insects

A *P. xylostella* population was sourced from the Institute of Vegetable and Flower Research, Chinese Academy of Agricultural Sciences (Haidian, Beijing). DBM individuals were reared using the vermiculite radish seedling method [27]. In brief, one-day-old radish seedlings were exposed to DBM adults for oviposition and replaced on a daily basis. Plantlets with DBM eggs were kept in rearing chambers, and the emerging larvae were allowed to feed upon 3–4-day-old radish seedlings until larval pupation. Next, pupae were collected and transferred to new adult rearing cages, where the emerging adults were allowed to mate and oviposit. Adults were provided with 10% honey water. Radish seedlings and moths were kept in 35 cm × 35 cm × 35 cm nylon screen cages and maintained at 27 ± 1 °C and 16:8 (L:D) photoperiod.

DBM pupae were collected and individualized in a ventilated 1.5 mL centrifuge tube. Newly emerged adults were selected, the sex distinguished, and placed in gauze-covered plastic cups (240 mL). For sex determination, morphological characteristics were used, i.e., the cylindrical-shaped abdomen of females and claspers of males. Supplemental nutrition was offered by supplying cotton balls dipped in 10% honey water. To determine the effects of temperature and relative humidity (RH) on WBF, 3-day-old unmated adults were used. To assess the effects of adult age and sex, we tested 1-, 3-, 5-, and 7-day-old adult males and females. Lastly, to evaluate the effect of feeding status, we either provided 3-day-old unmated adults with honey water or with purified water as a control.

### 2.2. Experimental WBF Assessment

Prior to measuring the WBF, each adult individual was CO_2_ anesthetized in order to attach it to an insect pin. Carbon dioxide was administered by directing the CO_2_ flow from a gas cylinder using a pressure regulator, a gas plate with minute exit holes, and a gas injector or syringe. We subjected each individual moth to CO_2_ at a flow rate of 2–5 L/min for 5–7 s until the moth ceased movement. Next, the moth was exposed to airflow to release CO_2_, and the scales of its dorsal thorax were gently removed using a brush. A small droplet of hot melt glue (DL5041 type, Deli, Ningbo, Zhejiang, China) was then applied to the tip of an insect needle, and this was pressed onto the dorsal thorax of the moth. After inserting this needle into a foam plate, the suspended DBM moth was transferred to a climate-controlled chamber for further experimentation. A stroboscope (pbx type, Monarch, Amherst, NH, America) was used to measure the WBF of each moth. Specifically, we adjusted the flash frequency of the stroboscope to be synchronized with the insect’s WBF, creating a visual perception in which its wings move slowly or approach stillness [28]. Once the wings thus approached stillness, the stroboscope reading was taken. For each individual and experimental condition, three different stroboscope readings were made.

To either test the effect of temperature or RH, experiments were conducted in a walk-in climate-controlled chamber. To assess the effects of RH, WBF was measured at a temperature of 25 ± 1 °C and variable relative humidity of 15%, 30%, 45%, 60%, 75%, and 90%. To assess the effects of temperature, WBF was measured at a fixed 45% ± 5% RH and variable temperature, i.e., 10 °C, 15 °C, 20 °C, 25 °C, 30 °C, and 35 °C. Prior to the onset of the experiment, moths were allowed to adapt to the experimental temperature or humidity for at least 10 min. About 20–30 males and females, i.e., a total of 40–60 individuals, were exposed to each experimental treatment.

To test the effects of age and nutritional status, WBF recordings were made in a climate-controlled chamber at 25 ± 1 °C and 45% RH. Prior to the onset of the experiment, moths were allowed to adapt to the experimental conditions for at least 10 min. About 30 male and 30 female insects, i.e., a total of 40–60 individuals, were tested under each of the four age groups or two feeding treatments.

### 2.3. Statistical Analysis

One-way ANOVA was used to analyze the effects of temperature, humidity, and age on *P. xylostella* WBF. A *t*-test was used to analyze WBF differences between individuals exposed to different adult nutrition treatments or between males and females under different conditions. We used binomial regression analysis to fit the effect of temperature on WBF. For this analysis, we used all data from a mix of both male and female moths, except for the data that were used to distinguish sex-related effects. All data were processed, analyzed, and plotted using GraphPad Prism 7.0 software.

## 3. Results

### 3.1. Temperature and Relative Humidity

Using negative binomial regression, we confirmed that *P. xylostella* WBF increases with temperature over a 10–35 °C range (Y = −0.02823X^2^ + 2.026X + 20.99; R^2^ = 0.4757). At 25 °C, 30 °C, and 35 °C, the highest wingbeat frequency attained a respective 52.95 Hz, 56.63 Hz, and 57.47 Hz. At 20℃, the WBF was significantly lower than at 30 °C (*p* = 0.0048) and 35 °C (*p* = 0.0011). Equally, the WBF at 15 °C was lower than at 20 °C (*p* < 0.05) and at 25 °C, 30 °C, and 35 °C (*p* < 0.001). Lastly, at 10 °C, the WBF readings were markedly lower than at any other temperature (*p* < 0.001) (Figure 1).

When exposed to varying relative humidity, the WBF showed an overall downward yet non-significant trend, with increasing humidity. The highest WBF was 56.05 Hz at 15% RH, and the lowest WBF was 51.32 Hz at 90% RH (Figure 2).

### 3.2. Age and Sex

Age did not affect the WBF of *P. xylostella*. The highest and lowest WBF values were recorded for individuals at 7 and 3 days of age, respectively (Figure 3).

In contrast, WBF differed between female and male individuals across temperature regimes (*p* < 0.001; Figure 4A), humidity regimes (*p* < 0.001; Figure 4B), age groups (*p* < 0.001; Figure 4C), and nutritional regimes (honey fed: *p* < 0.001; water fed: *p* < 0.01; Figure 4D). In addition, the females exhibit a higher WBF at 90% than at 15% RH (*p* < 0.05, Figure 4B), and, following honey consumption, the males in particular improved their WBF (*p* < 0.05, Figure 4D).

### 3.3. Adult Nutrition

Adult feeding status affected DBM wingbeat frequency (*p* = 0.0030), with honey-fed moths exhibiting a markedly higher WBF (54.66 Hz) than unfed individuals (49.52 Hz; Figure 5).

## 4. Discussion

Insect migration is shaped by a wide range of environmental and physiological factors, and the flight performance of an individual closely relates to its wingbeat frequency (WBF). As each insect species possesses a unique, distinguishable WBF, this feature can be used to remotely identify air-borne insects [18] and is also a decisive factor in flight aerodynamics research. Previously, WBF was measured by recording the sound waves that were generated during flight [20] or using high-speed cameras [21]. Our work confirms that, by adjusting the flash frequency of a stroboscope, one can quickly and accurately assess the WBF of small moths such as DBM. Further, their simple operation and relatively cheap price have made stroboscopes the tool of choice to measure WBF in various insects [17,26]. Through this method, we clarified that temperature, sex, and adult nutrition greatly determine the WBF of a globally damaging pest, i.e., the diamondback moth *P. xylostella*. By thus filling a critical knowledge gap on *P. xylostella* WBF, our work provides invaluable guidance for future flight biology research on this pest.

Climatic conditions exerted variable effects on the wingbeat frequency of *P. xylostella*. Over a 10–35 °C range, the WBF of both male and female adults increased with temperature, although the former attained their absolute maximum at 30 °C. These patterns are reminiscent of those of other lepidopterans, e.g., *Ctenoplusia agnata* and *Spodoptera litura* [26]. For *Helicoverpa armigera* at 13–30 °C and *Barathra brassicae* at 12–32 °C, WBF also increases with temperature to then (slightly) decline at the uppermost bracket [29,30]. Hence, though the exact maxima are likely defined by species-specific developmental or survival thresholds, the overall WBF response mode exhibits similarities across the (few) studied Lepidoptera. Relative humidity exerted no effects on WBF overall, though sex-specific patterns reveal how the WBF of *P. xylostella* females steadily declines with humidity and attains its absolute maximum at 15% RH (Figure 4B). This pattern is similar to that of other (non-lepidopteran) insects such as *Agrotis ypsilon* and *Periplaneta americana* [26]. For instance, *P. americana* males exhibit a higher WBF at 95% than at 50% RH [31]. These differences between sexes of a given species may be attributed to morphological aspects, i.e., increased humidity may disproportionately affect the larger female moths.

Under identical experimental conditions, adult males consistently exhibited a higher WBF than females. This is in line with findings for other noctuids such as *Agrotis ypsilon*, *Spodoptera litura*, and *Spodoptera exigua* but different from the crambid *Cnaphalocrocis medinalis* and the cockroach *P. Americana* [25,26,31]. As above, sex-specific differences may be attributed to the generally larger body size of female Lepidoptera. Hence, females possibly require slower wing movements to ensure sufficient lift and speed. Further, nutritional status affected the overall *P. xylostella* WBF and previous carbohydrate feeding raised it by 10%. Obviously, the intake of carbohydrate-rich foods such as honey provides energy and thus favors (migratory) flight. Following honey consumption, DBM males in particular improved their WBF (Figure 4D). Differences between sexes are related to their varying energy metabolism and mirrored in-flight performance. For instance, earlier work shows how 3-day-old *P. xylostella* females are able to fly longer distances than male moths [15]. Females thus likely store enough energy for sustained (migratory) flight, while males may rely more upon frequent carbohydrate consumption. Lastly, age neither affected *P. xylostella* WBF overall nor sex-specific WBF patterns. This trend differs from that of *C. medinalis*, in which WBF changes with adult age [25].

In China and abroad, DBM migration has been extensively researched. As *P. xylostella* outbreaks were regularly reported in remote, isolated areas outside the species’ overwintering range, the species was assumed to engage in long-distance migration. This was subsequently confirmed through high-altitude capture, radar monitoring, and population genetics [14,32]. This work has proceeded in parallel to flight research. Tethered flight trials revealed how adult age, food, and mating status affected flight performance [15], while other studies showed how flight performance is also modulated by larval-rearing hosts, i.e., by impacting adult morphology [33]. In recent years, trajectory simulation (or pathway analysis) and insect radar detection have been increasingly used in insect migration studies [13,19,29]. The validity and reliability of trajectory analysis depend upon the correct detection of (multiple) flight variables, i.e., individual flight performance, seasonally variable flight altitude, and the prevailing temperature and wind speed within the actual flight corridor [32]. The tactical use of insect radar can help to characterize several of these parameters and thus improve trajectory simulation. Indeed, insect radar can record the body length, body width, mass, flight height, and WBF of flying insects. Radar-derived flight parameters such as WBF match the ones measured in the laboratory [34]. As such, by ascertaining *P. xylostella* WBF under specific climatic conditions in the laboratory and pairing those values with radar readings, one can more accurately determine flight height and improve the overall accuracy of trajectory simulation.

Overall, our work showed how the WBF of *P. xylostella* is modulated by ambient temperature, adult nutrition, and sex. Over the past few decades, diamondback moth-related issues have been progressively aggravating—which is ascribed to its high levels of insecticide resistance [1], agricultural intensification, and global warming. In particular, climate-driven range expansion and amplification of the migratory range lie at the core of the increasingly severe DBM issues in northern China. Thus, research on *P. xylostella* migration and its environmental determinants urgently needs to be strengthened. One equally needs to clarify how other variables, e.g., mating status, wind speed, atmospheric pressure, and sex-specific (molecular, physiological) mechanisms shape flight performance. Trajectory analysis, tactically paired with radar monitoring and laboratory assays, carries a lot of promise to provide timely, point-specific guidance for DBM preventative or curative interventions [35,36]. More accurate outbreak forecasts in principle can counter insecticide abuse, reduce agricultural losses and, in the end, favor the uptake of a wide array of environmentally sound technologies such as biological control.

## Figures and Tables

**Figure 1 insects-15-00138-f001:**
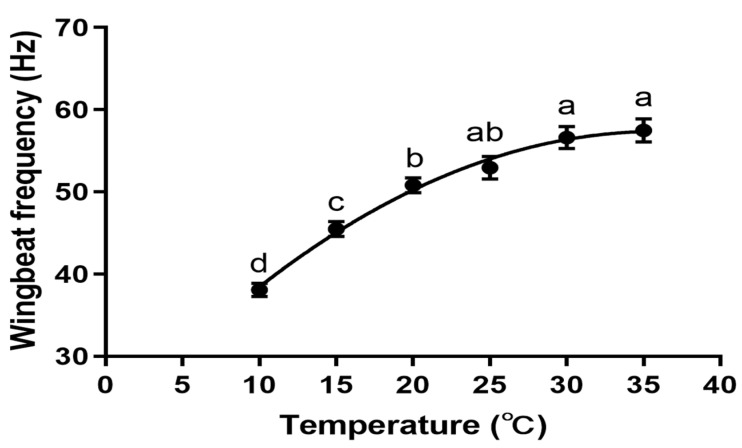
Effect of temperature on the WBF of *P. xylostella*. One-way ANOVA, F _(5, 226)_ = 41.57, *p* < 0.001, different letters above the error bars indicate statistically significant differences in WBF (*p* < 0.05, with Tukey’s multiple comparisons test). *n* (10 °C, 15 °C, 20 °C, 25 °C) = 40, *n* (30 °C) = 38, *n* (35 °C) = 34. The data were collected from a mix of both male and female moths.

**Figure 2 insects-15-00138-f002:**
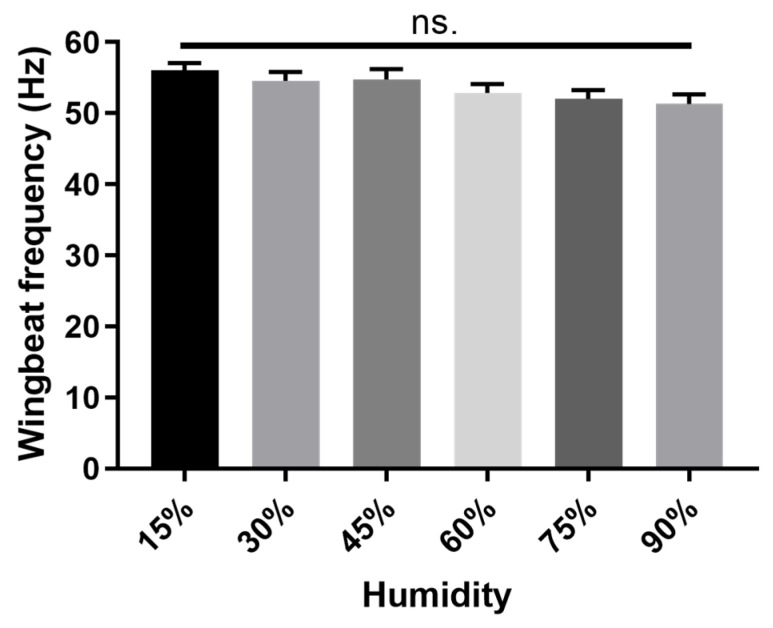
Effect of humidity on the WBF of *P. xylostella.* One-way ANOVA, F _(5, 233)_ = 2.047, *p* = 0.0729, no significant differences (‘ns’) were recorded across experimental treatments (*p* > 0.05 with Tukey’s multiple comparisons test). *n* (15%, 30%, 45%, 60%, 75% RH) = 40, *n* (90% RH) = 39. The data were collected from a mix of both male and female moths.

**Figure 3 insects-15-00138-f003:**
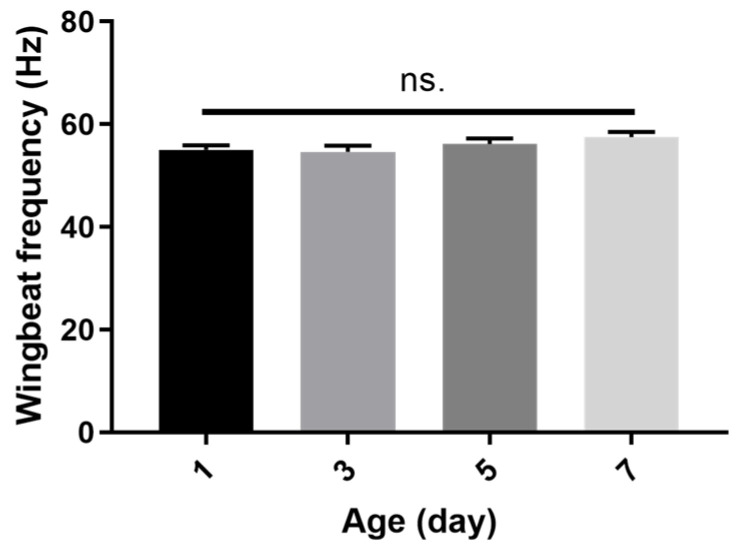
WBF for different ages of *P. xylostella*. One-way ANOVA, F _(3, 236)_ = 1.552, *p* = 0.2019, no significant differences (‘ns’) were recorded across age groups (*p* > 0.05, with Tukey’s multiple comparisons test). *n* = 60 in each group. The data were collected from a mix of both male and female moths.

**Figure 4 insects-15-00138-f004:**
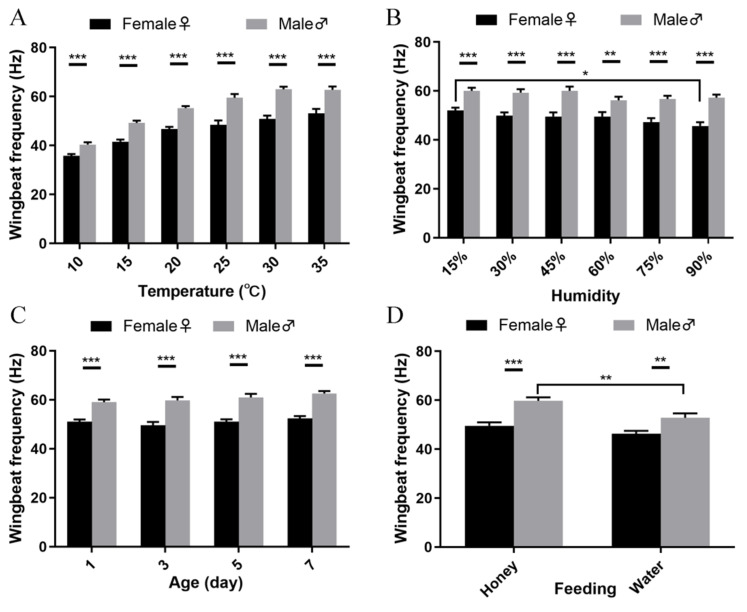
Sex-specific differences in *P. xylostella* WBF under various experimental regimes. Patterns are plotted for male and female WBF under different temperatures (**A**), humidities (**B**), different age groups (**C**), and under various feeding regimes (**D**). ‘**’ indicates a significant difference at *p* < 0.01, while “***” indicates a difference at *p* < 0.001 in the comparison between the male and female, by an unpaired *t*-test. ‘*’ in (**B**,**D**) indicates a significant difference at *p* < 0.05, by one-way ANOVA, F _(5, 114)_ = 2.081, *p* = 0.0728, *p* < 0.05 with Tukey’s multiple comparisons test in (**B**); F _(3, 120)_ = 15.20, *p* < 0.0001, *p* < 0.05 with Tukey’s multiple comparisons test in (**D**). *n* = approx. 20 each group in (**A**) and (**B**), *n* = approx. 30 for each group in (**C**,**D**). The black and grey bars present the data, which were collected from the female and male, respectively.

**Figure 5 insects-15-00138-f005:**
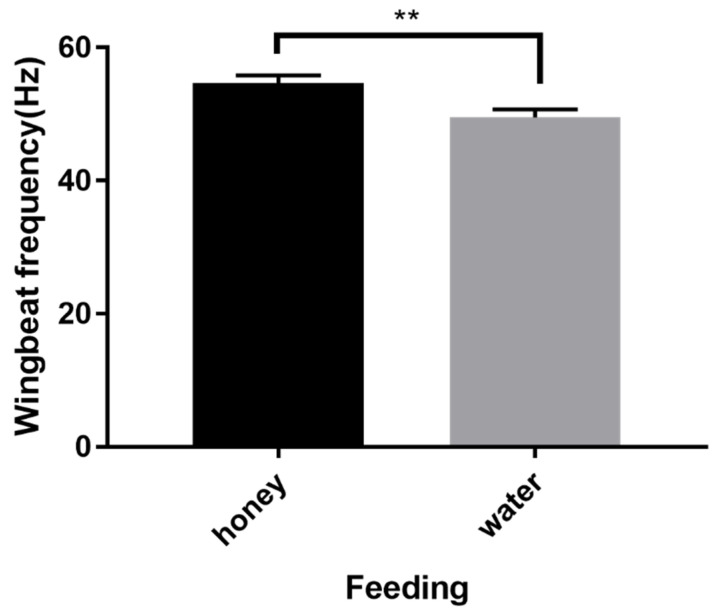
Effect of feeding regime on the WBF of *P. xylostella*. ‘**’ indicates a significant difference in wingbeat frequencies of *P. xylostella* (unpaired *t* test, *p* < 0.01). *n* = 60 for each group. The data were collected from a mix of both male and female moths.

## Data Availability

The original contributions presented in the study are included in the article, further inquiries can be directed to the corresponding author.

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
