# Peer review of "Nutritional Status, Sex, and Ambient Temperature Modulate the Wingbeat Frequency of the Diamondback Moth Plutella xylostella"

_insects, 2024, doi:10.3390/insects15020138_

Round 1

Reviewer 1 Report (Previous Reviewer 1)

Comments and Suggestions for Authors

There are still quite a lot of errors in the text of the paper – commented pdf attached.  I still don’t think the authors have really explained why the work is important – how knowledge about WBF will be used and is useful.  I think that the research needs to be clearly explained (improved English) and justified before it can be published.

Comments on the Quality of English Language

As above.  The English needs to be greatly improved in places.  

Author Response

There are still quite a lot of errors in the text of the paper – commented pdf attached.  I still don’t think the authors have really explained why the work is important – how knowledge about WBF will be used and is useful.  I think that the research needs to be clearly explained (improved English) and justified before it can be published.

Response: We are grateful to reviewer #1 for these extra valuable comments and suggestions. We have now critically revised the entire manuscript and rewritten large sections of text. We have attempted to better clarify the value of our research and have drastically improved English writing. The closing paragraphs of the Discussion section now further emphasize the value and importance of our research e.g., for pathway analysis / population forecasting and the ultimate implementation of environmentally-friendly pest management solutions.

1. Line 38: ‘reproduced’ -- should be reproduces not reproduced

Response: Accepted. Line 37: ‘Reproduced’ has been replaced with ‘reproduces’.

2. Line 57: ‘who’ -- which rather than who 

Response: Accepted. Line 57-58: The sentence ‘Throughout the year, the population of moths that migrate northward is larger than those who migrate southward’ has been revised in the updated version.

3. Line 85: ‘placed in’ -- placed with not in

Response: Accepted. Line 89-92: The sentence has been revised in the updated version.

4. Line 87: ‘pupa was’ -- pupae were 

Response: Accepted. Line 92: ‘pupa was’ has been replaced with ’pupae were’.

5. Line 87: ‘placed in’ -- with instead of in the 

Response: Line 88-90: The expression has been revised in the updated version.

6. Line 88: ‘seedling’-- seedlings

Response: Accepted. Line 94: ‘Seedling’ has been replaced with ‘seedlings’.

7. Line 89: ‘sown for 3-4 days’ -- sown 3-4 days previously 

Response: Accepted. Line 90-91: The expression has been revised in the updated version.

8. Line 90: ‘The above radish seedlings’ -- above not needed

Response: Accepted. Line 94-95: The sentence has been revised in the updated version.

9. Line 91: ‘raised in a 35cm*35cm*35cm nylon net’ -- were raised in 35cm... nylon nets. 

Response: Accepted. Line 94-95: The sentence has been revised in the updated version.

10. Line 91: ‘The feeding conditions’ -- this is temperature not feeding conditions

Response: Accepted. Line 95: The expression has been revised in the updated version.

11. Line 94: ‘the tube was perforated to prevent the death of the test insects’--from what? 

Response: Line 96: we now clarify that the ’insects were individualized in a ventilated tube’. Evidently, ventilation holes are provided so the insect does not die from suffocation.

12. Line 96: ‘observed’ -- observe 

Response: Accepted. Line 98-100:The expression has been revised in the updated version.

13. Line 96: ’terminal abdomen’ -- end of the abdomen?

Response: Accepted. Line 98-100:The expression has been revised in the updated version.

14. Line 97: ‘terminal’ -- end of the 

Response: Accepted. Line 98-100: The expression has been revised in the updated version.

15. Line 100: ‘unmatched’ -- unmated?

Response: Accepted. Line 102: ‘unmatched’ has been replaced with ‘unmated’ .

16. Line 100: ‘Adult male and female insects that emerged at 1, 3, 5, and 7 days were selected......’-- after what? Do you mean previously? 

Response: Accepted. Line 102-103: ‘Adult male and female insects that emerged at 1, 3, 5, and 7 days were selected......’ has been replaced with ‘To assess the effects of adult age and sex, we tested 1-, 3-, 5- and 7-day old adult males and females.’.

17. Line 102: ‘unmatched’ -- unmated? 

Response: Accepted. Line 104:‘unmatched’ has been replaced with ‘unmated’.

18. Line 107-108: ‘Before measuring the WBF, the moth was placed in a culture box at different temperatures for at least 10 minutes to adapt to the different environments.’ -- each temperature rather than different temperatures?

Response: Accepted. Line 122-129: The expression has been revised in the updated version.

19. Line 124: ‘relative temperature’ -- relative not needed 

Response: Accepted. Line 130-131: The sentence has been revised in the updated version.

20. Line 126: ‘the different environments’ -- environment 

Response: Accepted. Line 132: ‘environments’ has been replaced with ‘environment’.

21. Line 128-138: ‘The measurement method is as follows: Before measuring the WBF, the adult needs to be attached to the insect needle. Put the individual insect in a 10mL centrifuge tube and inject CO2 to anesthetize it. The CO2 narcosis equipment is constructed by a CO2 bottom, a structure to limit the gas pressure, a broad with countless pores and a pressure typed gas injector (syringe). The flow rate of CO2 we used to anesthetize the moth was around 2-5 L/min for 5-7 sec until the moth ceases movement. Once the insect ceases movement, quickly take out the test insect and put it on the air plate that can release CO2. Use a brush to gently sweep the scales on the back plate of the front chest of the moth. Dip a little hot melt glue on the head of the insect needle, stick the insect needle vertically into the dorsal thorax of the diamondback moth, insert the insect needle into the foam plate, and put the foam plate into the incubator to measure the WBF. Use a stroboscope (pbx type, Monarch) to measure the WBF of the moth. ’ -- poor English. 

Response: Line 107-121: The entire paragraph has been rebuilt and the poor English remediated.

22.Line 150: ‘The binomial was used to fit the effect......’ -- binomial what? 

Response: Accepted. Line 140: ‘binomial’ has been replaced with ‘binomial regression analysis’.

23. Line 157: ‘The results showed that there was a negative binomial regression relationship between temperature and the WBF of P. xylostella.’ -- should this be the other way round? 

Response: Accepted. Line 146-152: This entire paragraph has been corrected and rebuilt.

24. Line 169: ‘among’ -- between 

Response: Accepted. Line 154-157: The caption of Figure 1 has been corrected and rebuilt.

25. Line 190: ‘regimens’ --regimes 

Response: Accepted. Line 174: ‘regimens’ has been replaced by ‘regimes’.

26. Line 198: ‘For different day ages’ -- day not needed 

Response: Accepted. Line 180-185: This entire paragraph has been corrected and rebuilt.

27. Line 207: ‘regimen’ -- regime 

Response: Accepted. Line 191: ‘feeding regimen’ has been replaced by ‘feeding regime’.

28. Line 214: ‘Early insect WBF’ -- not expressed well! 

Response: Accepted. Line 198: ‘Early’ has been replaced by ‘Previously’.

29. Line 215-216: ‘wing frequency’ -- beat? 

Response: Accepted. Line 198-199: The sentence has been revised in the updated version.

30. Line 216: reference? 

Response: Accepted. Line 199: The reference ‘21’ has been added.

31. Line 217: ‘but it has high equipment requirements’ -- not sure what you mean. 

Response: This sub-sentence has now been removed from the revised manuscript.

32. Line 223-225: ‘These data fill the gap in the research on the WBF in this species to provide a data reference for the flight biology research of P. xylostella.’--not well expressed. 

Response: Accepted. Line 205-206: The sub-sentence has been replaced with ‘By thus filling a critical knowledge gap on P. xylostella WBF, our work provides invaluable guidance for future flight biology research on this pest.

33. Line 228-229: ‘The WBF of the small- and medium-sized diamondback moths in this study at different temperatures is similar to that of Ctenoplusia agnata and Spodoptera litura’ -- did you do any work on moth size?

Response: We didn’t do any work on moth size. And the expression has been revised and rebuilt in the updated version (Line 209-211).

34. Line 233: ‘insects’ -- delete 

Response: Accepted. Line 215: ‘insects’ has been deleted.

35. Line 243-248: ‘Although a different result was shown in our experiment, an analysis by sex revealed a significant impact on females, with their WBF showing an overall downward trend. The WBF of female moths is highest under 15% relative humidity (RH) conditions, significantly higher than that under 90% RH conditions. The WBF is not significantly affected under other humidity conditions. This may be because the larger size of the female moth increases the impact of humidity on WBF.’--which experiment? 

Response: Data about this experiment have now been added in the Figure 4B.

36. Line 250: ‘day’ -- delete 

Response: Accepted. Line 237-239: The expression has been revised and rebuilt in the updated version.

37. Line 254-257: ‘The WBF of female adults was highest at 7 days of age and lowest at 3 days of age (Figure 4C). The WBF of male adults was highest at 7 days old and lowest at 1 day old (Figure 4C). These differences may be because of the different development periods between the male and female moths.’--but not significant? 

Response: Yes, there is no significant difference between different age of both male and female moth. We further clarified this in the revised manuscript (Line 237-239).

38. Line 258:’Supplementing’ --supplementary

Response: Accepted. The expression has been revised in the updated version.

39. Line 258: ‘has’ --have 

Response: Accepted. The expression has been revised in the updated version.

40. Line 261-263: ‘but different phenotype were shown between the male and female moths. The WBF of the male insects fed honey water was significantly higher than for those fed purified water.’ --What about the females? 

Response: We have now revised this entire paragraph.

41. Line 266: ‘restore’ --correct word? 

Response: As suggested, Line 232-234: The sentence has been revised.

42. Line 269-277: ‘Besides the difference in the nutrition supplementation, a significant difference was found in the WBF between male and female adults of P. xylostella under the same conditions: the WBF of the males was higher than that of the females, and the differ-ence was extremely significant (Figure 4). This is similar to findings for Agrotis ypsilon, Spodoptera litura, and Spodoptera exigua of the Noctuidae family but different from Cnaphalocrocis medinalis and Periplaneta Americana 25, 28, 29. This situation may be related to the different body size between male and female moths in the order Lepidoptera, where females have a larger body size than the males, and for females, slower wing movements can provide sufficient lift and speed.’--Is this in the best place? 

Response: As suggested, we have now rebuilt the entire paragraph.

43. Line 278: ‘because it was used to be found in the areas out of its overwintering ability.’-- poor English. 

Response: The sentence has been revised in the updated version.

44. Line 285: ‘with’ -- on 

Response: Accepted. The expression has been revised in the updated version.

45. Line 290-291: ‘cannot be separated from accurate flight altitude’ --not sure what you mean. 

Response: The sentence has been revised in the updated version.

46. Line 295: ‘collected’ -- correct word? 

Response: Accepted. Line 252-256: The sentence has been revised in the updated version.

47. Line 307: ‘more’ --not needed. 

Response: Accepted. The entire section has been revised and rebuilt.

Reviewer 2 Report (Previous Reviewer 2)

Comments and Suggestions for Authors

The clarity of this paper has improved since the last submission. However, several issues remain unaddressed. Notably, the authors have only used one sentence in the results section of 3.3 but have presented some related results in the Discussion section. My major concern lies with the sample size. The authors mentioned that a total of 20 males and 20 females were tested under different humidities and temperatures, which is not sufficient.

My suggestions are as follows:

As mentioned in Lines 107-108, the authors tested a total of 40 moths—20 males and 20 females. For each temperature, they tested an average of 3.3 moths from each sex. This sample size is too small for the results to be statistically significant for publication. The same concern applies to the sample size for testing wingbeat frequencies under relative humidities.

Line 95: "The male has a grip device" – Could you be referring to claspers?

Line 133: Use the term "dorsal thorax" instead of "back plate of the front chest." This was a point I raised in my previous review.

Lines 182-183: The caption of Figure 3 is not legible.

Figure 5: Please specify whether the data presented were collected from both male and female moths, or solely from individuals of one gender. This was a concern in my last review that has not been addressed.

Lines 214-215: There is a reference missing.

Lines 259-262: Results on the WBF of males and females fed honey water versus those fed purified water are discussed but not presented in the Results section. There should be a detailed presentation of these findings in Section 3.3, not just a single sentence.

Line 301: A reference is needed after the statement "has a high level of resistance to pesticides."

Comments on the Quality of English Language

Line 94: The phrase "to observed" should be corrected for proper grammar.

Lines 228-230: Please review the sentence beginning with "Moreover," as it is currently difficult to understand.

Author Response

The clarity of this paper has improved since the last submission. However, several issues remain unaddressed. Notably, the authors have only used one sentence in the results section of 3.3 but have presented some related results in the Discussion section. My major concern lies with the sample size. The authors mentioned that a total of 20 males and 20 females were tested under different humidities and temperatures, which is not sufficient.

Response: We wish to thank reviewer #2 for his/her valuable comments. The new Figure 4 now covers some of the data that were not described in the earlier draft of our manuscript. We also wish to clarify that the sample size for each experimental treatment amounted to about 40-60 individuals i.e., 20-30 males and 20-30 females. We are confident that this sample size is sufficient to pick up meaningful effects of a given experimental treatment. To avoid confusion among future readers, we also clarify this better in the main text. 

My suggestions are as follows: 

1. As mentioned in Lines 107-108, the authors tested a total of 40 moths—20 males and 20 females. For each temperature, they tested an average of 3.3 moths from each sex. This sample size is too small for the results to be statistically significant for publication. The same concern applies to the sample size for testing wingbeat frequencies under relative humidities.

Response: We wish the clarify that around 20 males and 20 females were tested for each experimental treatment (Line: 133). This has now also been better clarified in the text. 

2. Line 95: "The male has a grip device" – Could you be referring to claspers? 

Response: Line 99: ‘a grip device’ has been replaced with ‘claspers’.

3. Line 133: Use the term "dorsal thorax" instead of "back plate of the front chest." This was a point I raised in my previous review.

Response: As suggested, Line 112&114: The ‘back plate of the front chest’ has been replaced with ‘dorsal thorax’.

4. Lines 182-183: The caption of Figure 3 is not legible.

Response: Accepted. It has been revised in the updated version (Line 169-171).

5. Figure 5: Please specify whether the data presented were collected from both male and female moths, or solely from individuals of one gender. This was a concern in my last review that has not been addressed. 

Response: All the data were collected from both male and female moths, except the data that was used to make the comparison between male and female (data of Figure 4). This has now been explained better in the updated version.

6. Lines 214-215: There is a reference missing. 

Response: We wish to thank reviewer #2 for picking up this important omission; reference ‘21’ has now been added (Line 199).

7. Lines 259-262: Results on the WBF of males and females fed honey water versus those fed purified water are discussed but not presented in the Results section. There should be a detailed presentation of these findings in Section 3.3, not just a single sentence. 

Response: Related data have now been added in the updated version (Figure 4D).

8. Line 301: A reference is needed after the statement "has a high level of resistance to pesticides."

Response: Accepted. Line 263: The references ‘1’ and ‘9’ have been added in the updated manuscript.

9. Line 94: The phrase "to observed" should be corrected for proper grammar.

Response: This has been revised in the updated version.

10. Lines 228-230: Please review the sentence beginning with "Moreover," as it is currently difficult to understand.

Response: This has been revised in the updated version (Line 211-212).

Round 2

Reviewer 1 Report (Previous Reviewer 1)

Comments and Suggestions for Authors

This manuscript is much improved.  I've made a few small suggestions about the English in the attached file.

Comments on the Quality of English Language

English is much improved.  Thank you.

Author Response

This manuscript is much improved.  I've made a few small suggestions about the English in the attached file.

Response: Many thanks for your suggestions about the English, and we accepted all.

Reviewer 2 Report (Previous Reviewer 2)

Comments and Suggestions for Authors

The authors addressed my questions well. I only have one comment.

I noticed the authors added the nutrition assay data of males and females to Figure 4. But still, in Figure 5 and Section 3.3, the authors should indicate whether the data presented were collected from a mix of male and female moths, or solely from individuals of one gender.

Author Response

I noticed the authors added the nutrition assay data of males and females to Figure 4. But still, in Figure 5 and Section 3.3, the authors should indicate whether the data presented were collected from a mix of male and female moths, or solely from individuals of one gender.

Response: Accepted. We have indicated that the data presented were collected from a mix of male and female moths in 2.3, Figure 1, 2, 3 & 5, respectively.

This manuscript is a resubmission of an earlier submission. The following is a list of the peer review reports and author responses from that submission.

Round 1

Reviewer 1 Report

Comments and Suggestions for Authors

This is a difficult paper to review because it is badly written and the English is poor.  I think it needs work on the English before it can be reviewed properly.  I have written some comments below.  I have made some more detailed comments on the Introduction as this section was written the best.  I have made general comments about the other sections. 

It is also based on a small number of experiments.   

In addition to my comments below I have some more general comments which relate to the Discussion but also to the way that the whole paper is written:

I think that the Discussion needs to compare the measurements of WBF in this study with any other measurements of WBF for P. xylostella.  It needs to critique the methodology compared with other methods used e.g. by Tercel et al (2018).  I am not convinced that it has really explained well why knowledge of WBF will be useful in future (there are now some insect detection/identification techniques using WBF I think (e.g. Kalfas, I., De Ketelaere, B., Beliën, T. and Saeys, W., 2022. Optical Identification of Fruitfly Species Based on Their Wingbeats Using Convolutional Neural Networks. Frontiers in Plant Science13, p.812506.)).  I am not convinced how the knowledge about the various factors and their effect on WBF will be used.  Will the temperature be measurable at the heights that P. xylostella migrate?

Abstract

Lines 16-19  I don’t think this really explains the situation very well.

Introduction

Lines 36-37  I think that the first sentence needs to be referenced.

Lines 37-38  There is a difference between the areas where it can complete its whole life-cycle (e.g. overwinter) and where it arrives as a migrant.

Lines 39-41 – not well explained.

Line 55 Throughout the year, they mainly migrate northward. How does this fit with the previous sentences?

Lines 74-76 – not well explained.

Materials and methods – English very poor – not well explained.

Results – also unclear in many places.

Discussion – English very poor.  No need to re-iterate the results at the start.

Comments on the Quality of English Language

As above.  The English is very poor and the paper needs work in this respect.

Reviewer 2 Report

Comments and Suggestions for Authors

This paper investigates the effects of temperature, humidity, age, and food supplementation on the wingbeat frequency of the Diamondback Moth using a stroboscope. The experimental design is sound, and the figures presented are effective. The conclusions drawn are in line with the evidence provided. However, the Discussion section requires further development. While the authors have summarized all results in the first paragraph, the comparisons made with other research in the second and third paragraphs could be structured more effectively. It may be more beneficial to focus on each result individually, followed by a discussion of the relevant literature, to allow for a clearer comparison and synthesis of the findings. The fourth and fifth paragraphs provide too much background information, which could be either condensed or moved to the Introduction. The implications and future directions in the Discussion may be expanded.

Specific comments:

Line 28-29, there is extra space between those two lines, please fix it.

Line  40, " with the southern provinces being more harmful"--The word "harmful" typically describes something that causes harm or damage, so it doesn't make sense to describe a location as "more harmful". I suggest the authors revise this sentence to clear up the ambiguity. For example, change it to "with its effects being more pronounced in the southern provinces."

Line 90, Could you provide method or citations for distinguishing the male and female adult moths?

In Line 116, where '10% honey water and water' is mentioned, could you please specify the type of water used? For instance, was distilled water or another form of purified water employed?

Line 123, do you mean you used a syringe for CO2 injection to the tube? If so, specify the detail of the syringe and the volume of CO2.

Line 123, "after the insect stops struggling" can be replaced with "Once the insect ceases movement" to sound more scientific and formal.

Lines 126-127, please use the term "dorsal thorax"  instead of "back plate of the front chest".

Line 172, remove the extra period after xylostella.

For Figure 5, could you please clarify whether the data presented were collected from both male and female moths, or from individuals of a single gender? This information is crucial for understanding the results accurately.

Lines 203-205, in the sentence 'Humidity has no significant effect on the WBF of adults. Specifically, different genders have an impact on the female adults of the moth, showing an overall downward trend.' is somewhat confusing and needs clarification. I have rewrote the sentence as follows "Humidity was found to have no significant effect on the WBF of adult moths. However, an analysis by gender revealed a significant impact on females, with their WBF showing an overall downward trend." Please check if this is correct.

Line 207, The phrase 'the age of the day' in the sentence 'This study found that the age of the day had no significant effect on the WBF of the moth.' is ambiguous. If you are referring to the daily age increments of the moths, please consider rephrasing to 'the age of the moth in days.'. I have also noted that you used day age in Line 208, please also rephrase this.

Lines 212-216.  There are some inconsistencies regarding the effects of honey water on the WBF of male and female insects. It seems that honey water increases WBF for both genders, but the effect is statistically significant only for males. Please clarify.

In Lines 199-219, where you are summarizing the results in the first paragraph of the discussion section, it would be beneficial to reference the corresponding figures for each result. This will help readers in locating the visual data that supports your findings. For instance, the sentence 'When the temperature is between 10 and 35°C, the WBF of female adult increases with the temperature rise.' could be enhanced by specifying the related figure, as in 'When the temperature is between 10 and 35°C, the WBF of female adults increases with the temperature rise (see Figure 1)."

In the fourth and fifth paragraphs of the Discussion, I found the background information to be too extensive. I would suggest that the authors condense the background information and focus more on discussing how the findings contribute to this area. Alternatively, consider moving some of the background information to the Introduction.

Comments on the Quality of English Language

Please review the manuscript for adherence to standard English conventions, particularly regarding the spacing between values and units. For instance, a space should be added to read '10% honey,' '6.7 Hz,' and '95% RH.' Extensive revisions may be required to ensure the clarity and readability of the text.